# Changes in unmet need for family planning among married women of reproductive age in Nigeria: A multilevel analysis of a ten-year DHS wave

**Funmilola Folasade Oyinlola**(iD)**\*, Joseph Ayodeji Kupoluyi, Olufemi Mayowa Adetutu**

Faculty of Social Sciences, Department of Demography and Social Statistics, Obafemi Awolowo University, Ile-Ife, Nigeria

\* foyinlola@cartafrica.org

**Data Availability Statement:** Data for this study is publicly available at the Demographic and Health Surveys (DHS) website (http://dhsprogram.com/

## Abstract

### Introduction

Unmet need for family planning [UNFP] remains a serious public health concern in Nigeria. Evidence suggests that UNFP remains high over the last fifteen years despite numerous policies and programmes aimed at generating demand for family planning. This study used three Demographic and Health Survey (DHS) conducted over a ten-year period (2008–2018) to assess the changes in unmet need for family planning and associated contextual determinants. Understanding changes in unmet need for family planning among women and its associated contextual factors is crucial for designing appropriate interventions.

### Methods

We analysed datasets the Nigeria Demographic and Health Surveys of 2008, 2013 and 2018 to assess changes and contextual determinants of unmet need for family planning. Data were analysed using frequency distribution, chi-square statistical test and multilevel binary logistic regression models. Due to the hierarchical structure of the data in which individuals are nested within households, multilevel mixed-effect logistic regression models were constructed. We used a multilevel binary logistic regression model after adjusting for variables not significant at the bivariate level. An adjusted odds ratio with 95% confidence interval was reported, with a p-value less than 0.05 declared to be significant predictors of unmet need for family planning.

### Results

Unmet need for family planning decreased from 20.21% to 16.10% between 2008 and 2013 but subsequently rose later from 16.10% to 18.89% between 2013 and 2018. The pattern of changes in unmet need for either limiting or spacing was consistently high over the 10-year period, with the highest rate of each of the indicators of unmet need for family planning occurring in 2018 while the lowest rate was in 2008, thus indicating an increase in the proportion of respondents having unmet need for family planning over the referenced period.

data/available-datasets.cfm). The data file, however, requires a written permission which will be approved by the ICF International.

**Funding:** The authors received no funding for this work.

**Competing interests:** The authors have declared that no competing interests exist.

Age of respondents, educational level, wealth status, religious affiliation, parity, sex of head of household, partner educational level, region of residence, and community socioeconomic status were significant factors associated with the unmet need for family planning across the different data waves in Nigeria ($p < 0.05$). An intraclass correlation (ICC) of 4.9% showed that the individual and household level factors had a greater influence on the variation in the unmet need for family planning than did community factors in Nigeria.

## Conclusion

The overall prevalence of unmet need for family planning was consistently high over the ten-year period and community-level factors had lowest influence on the variation in unmet need for family planning compared to household and individual-level factors in Nigeria. Policies and interventions should focus on improving women's socio-economic and demographic characteristics at individual, household, and community levels to improve unmet need for family planning.

## Introduction

Increased use of family planning is one crucial strategy for improving sexual and reproductive health of women across the globe. Unmet need for family planning refers to the magnitude of sexually active women who desire to stop or delay childbearing but are not employing any effective methods to prevent conception [1]. Women of reproductive age are a vulnerable group for adverse sexual and reproductive health (SRH) outcomes, including unintended pregnancy, unsafe abortion, sexually transmitted infections, pregnancy complications and maternal deaths [1]. Women continue to face health challenges and disproportionately bear a huge burden of adverse maternal and child health outcomes owing to non-use of contraception and unmet need for family planning [2].

In 2019, 220 million had an unmet need for contraception in the world [2]. African countries bear 20% of unmet need for family planning compared to 10% in the world [3]. A multi-country study in sub-Saharan Africa (SSA) reported 23.7% unmet need for family planning among women of reproductive age [4]. The study also found that age of women, educational status, age at marriage, knowledge of family planning, parity, number of under-five children and knowledge of modern contraception, household size, visit to a health facility, partner's education, place of residence, community education and region of residence as predictors of unmet need for family planning.

Unmet need for family planning in Nigeria aligns with other countries in sub-Saharan Africa. For instance, unmet need for family planning oscillates between 14% and 20% in the past two decades in Nigeria [5]. Similarly, total fertility rate (TFR) remains high at 5.5 children per woman, with a higher sub-national rate, 7.7, in some Northern parts of Nigeria [6]. This has a high propensity for a sustained rapid population growth in Nigeria. Other studies conducted in Ghana among married women reported high prevalence of unmet need for family planning, as well as a range of socio-demographic and economic factors [7,8]. There is an avalanche of studies in Nigeria which established that unmet need for family planning is associated with negative health outcomes, such as unintended pregnancy, unsafe abortion and pregnancy complications [9–12]. While several interventions have been implemented for family planning demand generation in Nigeria [13–15], a vast majority of women who desire to

use family planning fail to use. Besides numerous benefits of contraception, modern contraceptive prevalence rate remains low at 18% in Nigeria [6], compared to some countries, especially the Southern bloc of sub-Saharan Africa, including Malawi, Rwanda and South Africa with modern contraceptive prevalence rate of 50% or higher [16].

Numerous studies have identified a range of socio-economic and demographic factors at individual, household and community levels associated with unmet need for family planning in Nigeria and other African countries. Knowledge of family planning, wealth quintile, partner's education, religion, number of living children, fear of side-effects, husband's disapproval, media exposure and place of residence were found to be significantly associated with unmet need for family planning [17–22]. However, unmet need for family planning remains consistently high in Nigeria despite many interventions, robust commitment and huge financial outlays invested by the government to generate demand for contraception at no cost, provide high-quality family planning services, as well as adequately planned logistics and contraceptive supplies. This is done in a bid to achieve the 36% modern contraceptive prevalence rate earmarked for 2025 in Nigeria [6] and monitor progress towards achieving the target 3.7 of the Sustainable Development Goals, which aimed to ensure universal access to sexual and reproductive healthcare services by 2030 [23–25].

There is a paucity of evidence on contextual factors associated with changes in unmet need for family planning among women of reproductive age in Nigeria. Assessing the changes and contextual determinants of unmet need for contraception would help policy makers and programme managers gauge the effectiveness of previous policies and future strategies to addressing the problem of unmet need for family planning. Hence, this study examined changes in unmet need for family planning and its associated factors among women of reproductive age in Nigeria over a ten-year period (2008–2018).

## Sample design

The study used secondary data from three rounds of the Nigeria Demographic and Health Survey (NDHS) conducted in 2008, 2013, and 2018 that had been collected through a cross-sectional survey design. A detailed explanation of the sample design of the DHS surveys has been published previously [6,26,27], and on the DHS website: https://www.dhsprogram.com

## Data source, study population and sample size

The individual recode (IR) dataset of the Nigeria Demographic and Health Survey (NDHS) conducted in 2008, 2013, and 2018 was used for this study. The IR file covered women aged between 15 and 49 years in the households located in the primary sampling units (PSUs) in the enumerated area. The NDHS collected nationally representative information on sociodemographic characteristics, fertility, family planning, and domestic violence, among other health-related issues. The 2008, 2013 and 2018 NDHSs are the fourth, fifth and the sixth rounds of surveys, respectively, in Nigeria. The data were accessed from the URL: https://www.dhsprogram.com/data/available-datasets.cfm. A detailed explanation of the sampling methods and selection of respondents has been published elsewhere [19,24,28]. Thus, this study used the weighted pooled sample size of 80,497 married women aged 15–49 years (2008–2018) comprising 23,578 married women from 2008, 27,830 married women from 2013 and 29,090 married women from 2018.

## Study setting

Nigeria has 36 states and a Federal Capital Territory (Abuja). Administratively, the country was grouped into six geopolitical zones and 774 constitutionally recognised local government

areas (LGAs). There are more than 250 ethnic groups among which Yoruba, Hausa/Fulani, and the Igbo are the dominant groups [18]. Nigeria was ranked the seventh and as the most populous country in the world and in Africa, respectively [28,29]. The estimated population was more than 206 million inhabitants in 2020 with a growth rate of approximately 2.5% annually [28]. Even though Nigeria's population is unevenly distributed across the country, it is expected to increase even more in the near future given the significant changes presently observed in age structures [28,30]. In terms of contraceptive use and unmet needs, the prevalence of contraceptive use (CPR) in recent nationally representative surveys was 17% and the prevalence of unmet need for family planning (FP) was at 18.9% (6.8% for limiting and 12.1% for spacing) [6].

## Exclusion and inclusion

The revised definition of unmet need for family planning [26,31] was applied to estimate total unmet need (spacing and limiting).

As shown in Fig 1, the currently married or in union women (15–49 years) who were left out of the study were as follows: (i) current use of contraception; (ii) women who are not using method of contraception, who are pregnant or amenorrheic but whose pregnancy or birth was intended or wanted; (iii) women who were not using method of contraception, were not pregnant or non-amenorrheic, or who were fecund but wanted to wait for more than 2 years; and (iv) women who are not using contraception, who are not pregnant or non-amenorrheic, and who are infecund.

Other currently married or in-union women of reproductive age (15–49 years) who were included in this study were as follows: (i) women who were not using contraception, who were pregnant or amenorrheic, and whose pregnancy or birth was mistimed or unwanted; and (ii) women who were not using contraception who were neither pregnant or non-amenorrheic, who are fecund but wanted to wait for at least 2 years or wanted no more no more children

## Handling of missing values

In DHS, special codes are used throughout the data file for certain responses. All unanswered questions were declared as missing data. Thus, in this study, missing values were declared as missing and were excluded from the analysis.

## Measurements of variables

**Outcome variable.** The outcome variable was unmet need for family planning. It was defined as the total sum of the unmet need for spacing and the unmet need for limiting. In this study, the variable was dichotomized and coded as '1' if the respondent reported having an unmet need for spacing and/or for limiting and '0' otherwise. For unmet need for spacing and limiting, currently married or in union women who were not using contraception were defined as having an unmet need for spacing were coded as '1' if they were (a) pregnant or postpartum amenorrhoeic but reported their pregnancy as mistimed and, (b) fecund but not pregnant or postpartum amenorrhoeic who wanted a child in the next $\geq 2$ years or who are yet to decide whether to have a child or when to have a child and '0' otherwise. Similarly, currently married or in union women who were not using contraception were classified as having an unmet need for limiting childbearing and coded as '1' if they were (a) pregnant or postpartum amenorrhoeic but reported their pregnancy as unwanted and (b) fecund but not pregnant or postpartum amenorrhoeic but wanted no more children and '0' otherwise.

**Explanatory variable.** The explanatory variables for this study were the individual-level (demographic and socioeconomic) characteristics and contextual factors. These variables were

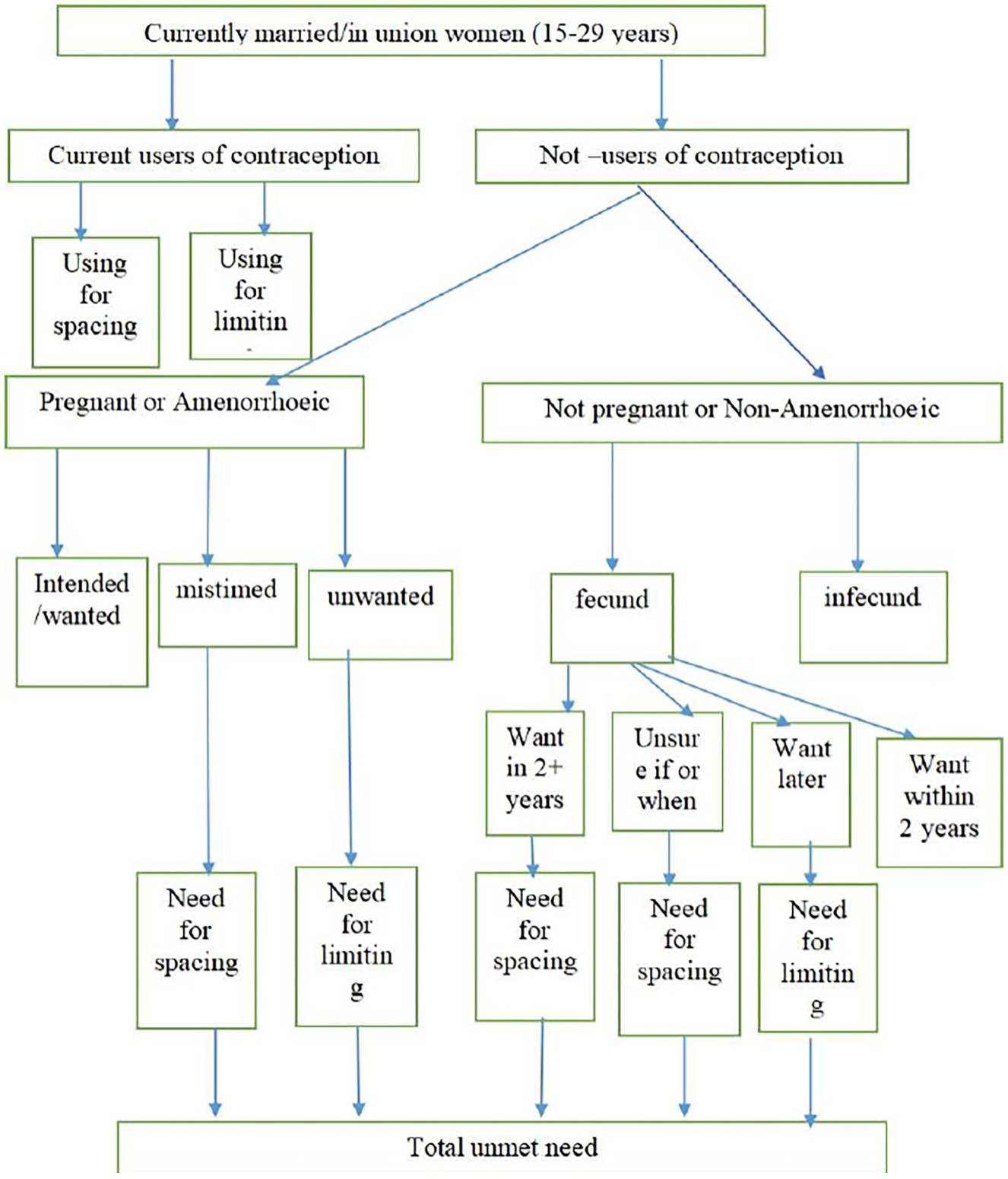

**Fig 1. Revised definition of unmet need for spacing and limiting.** Source: Bradley et al., 2012 [26].

selected based on what the literature considered important. The individual level variables were as follows: age group (15–19, 20–24, 25–29, 30–34, 35–39, 40–45 years); region (South West, South East, North West, North East, North Central, South South); place of residence (rural, urban); education (no formal education, primary, secondary, tertiary); wealth quintile (poorest, poorer, middle, richer, richest); religious affiliation (Christian, Islam and traditional/others); occupation (not working, working); parity (0, 1, 2–4, 5 and above); sex of head of household (male, female); partner's education (no formal education, primary, secondary, tertiary); ever experience child mortality (yes or no); and media exposure which was constructed by aggregating exposure to media [32].

A community-level or contextual factor is defined as the cluster of the Nigeria Demographic and Health Survey was also analysed in this study. Two community level variables were constructed by aggregating individual level characteristics at the cluster level as used in previous studies [33,34]. The median value was used as a cut-off point for the classification of the variables. These variables were: community socioeconomic status (low, moderate, high) and community knowledge of modern contraceptives (low, moderate, high).

## Statistical analysis

These three sets of DHS datasets were pooled and used to examine changes and contextual determinants of unmet family planning needs in Nigeria. The survey command (svyset [pw = wt], psu (mv021) strata (strata), which declared the data a survey data in STATA was applied to adjust for the cluster sampling techniques. The svy command used information on sampling weights, strata, and primary sampling units (PSUs) in the dataset to adjust for non-proportional allocation of the sample to strata and regions during the survey process and to account for the effect of over/under-sampling and nonresponse so as to restore the representative of the sample.

Descriptive statistics were used to describe the characteristics of the respondents using frequency distributions or percentage tables and charts for categorical variables. The mean and standard deviation for continuous variables were also computed. Inferential Statistics were used at the bivariate and multivariate levels on the pooled data to measure the association between explanatory variables and the outcome variable. The strength of association between the variables was measured using the Pearson chi-square test at the 95% level of significance. All significant variables with $p < 0.05$ at the bivariate level of analysis were included in the multivariate analysis. In addition, a multicollinearity test was performed on the predictor variables using a variance inflation factor (VIF) < 5 as the fixed point. To identify the influencing factors of the unmet need for family planning, multilevel mixed-effect logistic regression models were employed. This was done to account for the hierarchical nature of DHS data and to correct the estimated standard errors, which allow for the clustering of observations within units [35].

## Modelling

Models were run to estimate the fixed effects, random effects, and the goodness of fit. The 'xtmelogit' STATA command was used to estimate fixed effects and the random effect between cluster variations due to the hierarchical structure of the data in which individuals/women (level 1) are nested within households/communities (level 2).

## Fixed effects

In the modelling approach, four models were constructed: a null model (Model 0), which was fitted with no explanatory variable to test the random effect between-cluster variability in unmet needs for family planning, model 1, which included the outcome variable and the

individual/household-level variables, Model 2, which included the outcome variable and the community-level variables and Model 3, which was a combined model that included all variables (individual/household and community variables) to test for the net fixed and random effects. These fixed effects are reported as the odds ratios (ORs), 95% confidence intervals (CIs) and p values < 0.05.

### Random effects

The 'xtmelogit' and 'estat ic' STATA commands were used to obtain the variation in the unmet need for family planning across communities. The random effects (measure of variation) were measured using intraclass correlation (ICC), the proportional change in variance (PVC) and the median odds ratio (MOR). The ICC was used to evaluate the variation in unmet needs for family planning within or between clusters. It was calculated as: ICC% = $V_A$ / ($V_A$ + 3.29) × 100 where $V_A$ denotes the estimated variance of the clusters. The PCV was also used to measure the total variation attributed to individual-level factors and area-level factors. It was calculated as: PCV% = [($V_A$−$V_B$) / $V_A$] × 100, where $V_A$ is the variance in women's unmet need for family planning in the empty model and $V_B$ is variance in the successive models. The MOR is the median value of the odds ratio between the area at highest risk and the area at the lowest risk when randomly selecting two areas. It measures the increased risk that (in median) would have if moving to another area with a higher risk. This study showed the extent to which the individuals' probability of having unmet needs for family planning was determined by their residential area. It was calculated as: MOR = [exp. [$\sqrt{(2 \times V_A)} \times 0.6745$] ≈ exp. ($0.95\sqrt{V_A}$)

### Goodness of fit

The goodness of fit of the regression models was determined by using; (i) the variance inflation factor (VIF < 10) to assess multicollinearity, and (ii) the Akaike information criterion (AIC). A lower value of AIC value indicates a better fit of the model [36].

### Ethical approval and consent to participate

DHS is a secondary data which had already addressed ethical considerations. However, permission to use the NDHS data was sought and granted by Measuredhs. However, details regarding the data and ethical approval are available at: http://googl/ny8T6X

## Results

### Univariate results

Table 1 shows the socioeconomic characteristics of the respondents for all the years combined and for each NDHS year. Approximately one-fifth of the respondents (21.2%) were aged 25–29 years, while one-third of the respondents (33.6%) were from the northwestern zone of the country. Additionally, more than three-fifths of the respondents (63.6%) were living in rural areas, 46.6% of the respondents had no formal education, and 34.7% had secondary or tertiary education. With regard to wealth, more than one–fifth of the respondents were either the poorest households (22.2%) or poorer households (21.5%). Approximately two-fifths (40.4%) and nearly two-thirds of the respondents (58.5%) were either Christians or Muslims respectively. While the majority of the households (91.5%) were headed by a male, approximately 69.3% of the respondents were working. With regard to media exposure, slightly more than one-third of the respondents (36.2%) were exposed. Additionally, slightly more than one-third of the respondents' partners (37.8%) had no formal education. While the majority of the

**Table 1. Sociodemographic characteristics of currently married women (15–49) and prevalence of unmet need for family planning, Nigeria DHS 2008–2018.**

| Characteristics | Unmet need | | | |
|---|---|---|---|---|
| | **2008** | **2013** | **2018** | **Pooled (2008–2018)** |
| | N (%) 23578 (29.29) | N (%) 27830 (34.57) | N (%) 29090 (36.14) | N (%) 80497 (100) |
| **Age (Mean/±SD)** | **31.11 (±8.82)** | **31.55 (8.85)** | **31.89 (±8.63)** | **31.54 (8.77)** |
| 15–19 | 1863 (7.90) | 2251 (8.09) | 1927 (6.62) | 6041 (7.50) |
| 20–24 | 3659 (15.52) | 4362 (15.67) | 4362 (15.00) | 12383 (15.38) |
| 25–29 | 5112 (21.68) | 5913 (21.25) | 6060 (20.83) | 17085 (21.22) |
| 30–34 | 4173 (17.70) | 4869 (17.50) | 5417 (18.62) | 14458 (17.96) |
| 35–39 | 3575 (15.16) | 4302 (15.46) | 4841 (16.64) | 12719 (15.80) |
| 40–44 | 2711 (11.50) | 3226 (11.59) | 3457 (11.88) | 9394 (11.67) |
| 45–49 | 2484 (10.53) | 2907 (10.45) | 3026 (10.40) | 8417 (10.46) |
| **Region** | | | | |
| North Central | 3320 (14.08) | 3895 (14.00) | 4086 (14.05) | 11302 (14.04) |
| North East | 3585 (15.21) | 4679 (16.81) | 4841 (16.64) | 13106 (16.28) |
| North West | 7189 (30.49) | 10034 (36.06) | 9826 (33.78) | 27050 (33.60) |
| South East | 2139 (9.07) | 2333 (8.38) | 2893 (9.95) | 7365 (9.15) |
| South South | 2978 (12.63) | 2699 (9.70) | 2777 (9.55) | 8454 (10.50) |
| South West | 4366 (18.52) | 4189 (15.05) | 4666 (16.04) | 13221 (16.42) |
| **Place of residence** | | | | |
| Urban | 7375 (31.28) | 10124 (36.38) | 11790 (40.53) | 29289 (36.39) |
| Rural | 16203 (68.72) | 17705 (63.62) | 17299 (59.47) | 51208 (63.61) |
| **Education** | | | | |
| No formal education | 11120 (47.16) | 13470 (48.40) | 12955 (44.53) | 37546 (46.64) |
| Primary | 5143 (21.81) | 5336 (19.17) | 4580 (15.75) | 15059 (18.71) |
| Secondary | 5621 (23.84) | 6981 (25.08) | 8767 (30.14) | 21369 (26.55) |
| Tertiary | 1693 (7.18) | 2043 (7.34) | 2788 (9.58) | 6523 (8.10) |
| **Wealth quintile** | | | | |
| Poorest | 5408 (22.94) | 6424 (23.08) | 6008 (20.65) | 17840 (22.16) |
| Poorer | 5052 (21.43) | 5986 (21.51) | 6224 (21.40) | 17263 (21.45) |
| Middle | 4311 (18.28) | 4983 (17.91) | 5601 (19.26) | 14895 (18.50) |
| Richer | 4216 (17.88) | 5042 (18.12) | 5599 (19.25) | 14858 (18.46) |
| Richest | 4590 (19.47) | 5395 (19.38) | 5657 (19.45) | 15642 (19.43) |
| **Religion affiliation** | | | | |
| Christianity | 10281 (43.87) | 10581 (38.23) | 11558 (39.73) | 32421 (40.42) |
| Islam | 12753 (54.41) | 16812 (60.73) | 17375 (59.73) | 46939 (58.52) |
| Traditional/Others | 405 (1.73) | 287 (1.04) | 157 (0.54) | 849 (1.06) |
| **Parity** | | | | |
| 0 | 2402 (10.19) | 2823 (10.14) | 333 (8.02) | 7557 (9.39) |
| 1 | 3605 (15.29) | 4235 (15.22) | 4349 (14.95) | 12189 (15.14) |
| 2–4 | 10990 (46.61) | 12706 (45.66) | 13629 (46.85) | 37325 (46.37) |
| 5 and above | 6581 (27.91) | 8065 (28.98) | 8779 (30.18) | 23425 (29.10) |
| **Sex of Head of Household** | | | | |
| Male | 21393 (90.73) | 25449 (91.45) | 26773 (92.04) | 73616 (91.45) |
| Female | 2185 (9.27) | 2381 (8.55) | 2316 (7.96) | 6881 (8.55) |
| **Partner's educational** | | | | |
| No formal education | 9001 (38.79) | 10972 (39.79) | 12955 (44.53) | 30002 (37.79) |
| Primary | 4985 (21.48) | 5100 (18.50) | 4580 (15.75) | 14323 (18.04) |
| Secondary | 6315 (27.21) | 7731 (28.04) | 8767 (30.14) | 23778 (29.95) |
| Tertiary | 2906 (12.52) | 3769 (13.67) | 2788 (9.58) | 11287 (14.22) |
| **Experienced child death** | | | | |
| No | 14834 (62.92) | 18442 (66.27) | 20077 (69.02) | 53353 (66.28) |
| Yes | 8744 (37.08) | 9388 (33.73) | 9012 (30.98) | 27144 (33.72) |
| **Community socioeconomic status** | | | | |
| Low | 12827 (54.40) | 15050 (54.08) | 15682 (53.91) | 25146 (31.24) |
| Moderate | 1607 (6.81) | 3267 (11.74) | 2840 (9.76) | 25532 (31.72) |
| High | 9144 (38.78) | 9512 (34.18) | 10568 (36.33) | 29818 (37.04) |

(*Continued*)

**Table 1.** (Continued)

| Characteristics | Unmet need | | | |
|---|---|---|---|---|
| | **2008** | **2013** | **2018** | **Pooled (2008–2018)** |
| **Community knowledge of modern contraceptives** | | | | |
| Low | 7091 (30.07) | 9675 (34.76) | 8845 (30.41) | 24825 (30.84) |
| Medium | 7340 (31.13) | 18155 (65.24) | 20244 (69.59) | 25724 (31.96) |
| High | 9148 (38.80 | na | na | 29948 (37.20) |

na: Not available.

respondents (75.5%) have had two or more children, one-third of the respondents (33.7%) had experienced child death. For community literacy level and socio-economic status, more than one-third of the respondents (37%) resides in communities with high literacy level and high socioeconomic status.

### Trends in the prevalence of unmet need for family planning

Fig 2 shows the trends in the prevalence of unmet needs for family planning in Nigeria for each NDHS year. The results revealed an improvement in the percentage of respondents who had unmet needs for family planning as the percentage declined from 20.21% to 16.10% between year 2008 and 2013. However, the percentage of respondents who had unmet need for family planning increased from 16.10% in 2013 to 18.89% in 2018.

Fig 3 shows the trends in the prevalence of unmet needs for spacing and limiting for child-bearing in Nigeria for each NDHS years. The results show an improvement in the percentage of respondents who had unmet needs for limits, as the percentage dropped from 5.2% to

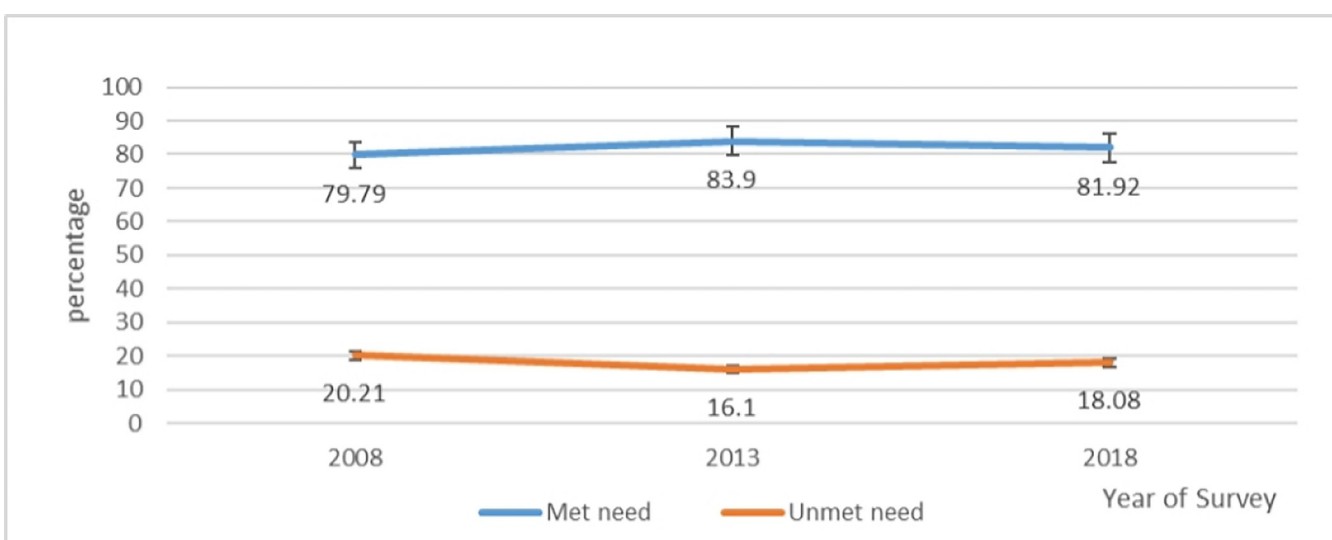

**Fig 2. Trends in the prevalence of unmet need for family planning in Nigeria, (NDHS 2008–2018).**

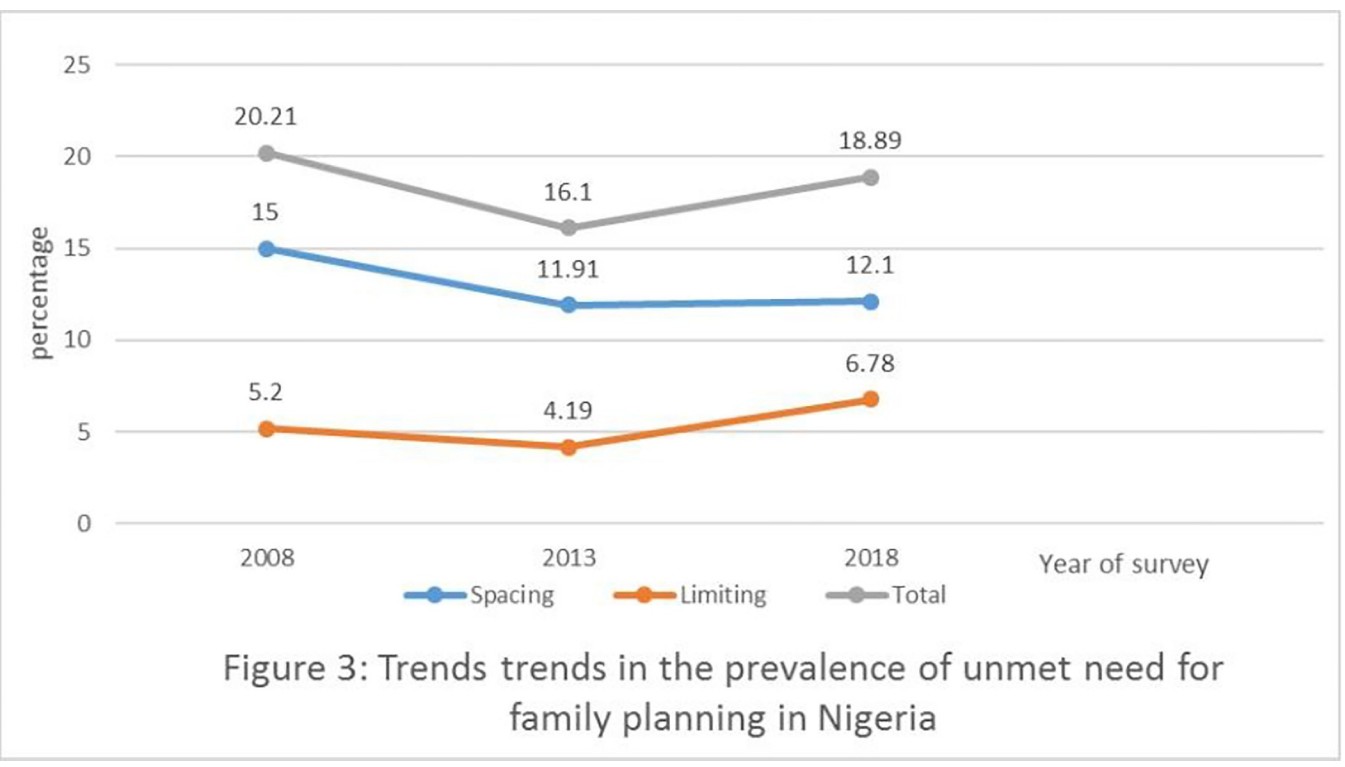

**Fig 3. Trends in the prevalence of unmet need for family planning in Nigeria.**

4.19% between 2008 and 2013. However, the percentage of respondents who had unmet need for limiting childbearing increased from 4.19 to 6.78% between year 2013 and 2018. Similarly, the trend in the prevalence of unmet needs for spacing shows that the percentage of respondents who had unmet need for spacing declined from 15% to 11.91% between 2008 and 2013. However, the percentage of individuals with unmet needs for spacing increased slightly from 11.91% to 12.10% between 2013 and 2018. Overall, the percentage of respondents who had unmet need for family planning dropped from 20.21% to 16.10% between 2008 and 2013 but later increased from 16.10% to 18.89% between 2013 and 2018.

## Changes in unmet need for family planning

Table 2 presents the proportional changes in unmet needs for family planning across the survey year and within the socio-demographic characteristics of married women in Nigeria (DHS 2008–2018). Overall, across the survey years, the result shows that the proportion of unmet need for family planning decreased by 20.33% from 20.12% to 16.10% between 2008 and 2013. Reasons for decline in unmet need for family by sociodemographic characteristics of the respondents' show that the variation is more pronounced among women from the North West, South East, South South and South West where level of unmet need for family planning dropped despite slight increase from the North Central and North East. The result also shows the contribution of married women aged 15–19, 20–24, 40–44 and 45–49 years to the decline in unmet need for family planning between 2008 and 2013. In addition, women from rural area, no formal education, primary and tertiary education level, as well as Christian and other religion except Islam contributed to the declined in unmet need for family planning during the period. A similar pattern of decline in unmet need for family planning is noticeable for

**Table 2. Proportional changes in unmet need for family planning across survey years and by socio-demographic characteristics.**

| Characteristics | Unmet need | | | |
|---|---|---|---|---|
| | **2008** | **2013** | **2018** | **Pooled (2008–2018)** |
| | N (%) | N (%) | N (%) | N (%) |
| Total | 4765 (20.21) | 4480 (16.10) | 5494 (18.89) | 9974 (12.39) |
| **Age** | | | | |
| 15–19 | 7.45 | 6.56 | 4.27 | 5.30 |
| 20–24 | 16.46 | 16.12 | 12.78 | 14.28 |
| 25–29 | 21.90 | 22.18 | 18.81 | 20.33 |
| 30–34 | 17.60 | 18.64 | 20.54 | 19.69 |
| 35–39 | 15.98 | 16.91 | 20.39 | 18.83 |
| 40–44 | 12.17 | 12.10 | 14.08 | 13.19 |
| 45–49 | 8.44 | 7.49 | 9.12 | 8.38 |
| **Region** | | | | |
| North Central | 13.04 | 20.42 | 15.25 | 17.57 |
| North East | 13.22 | 18.33 | 16.77 | 17.47 |
| North West | 31.37 | 26.94 | 25.71 | 26.26 |
| South East | 8.13 | 6.52 | 9.31 | 8.06 |
| South South | 16.22 | 13.35 | 14.06 | 13.74 |
| South West | 18.02 | 14.44 | 18.90 | 16.90 |
| **Place of residence** | | | | |
| Urban | 29.94 | 33.76 | 42.66 | 38.67 |
| Rural | 70.06 | 66.24 | 57.34 | 61.33 |
| **Education** | | | | |
| No formal education | 44.90 | 44.68 | 39.58 | 41.87 |
| Primary | 24.44 | 23.04 | 17.83 | 20.17 |
| Secondary | 25.46 | 26.93 | 33.88 | 30.76 |
| Tertiary | 5.19 | 5.34 | 8.70 | 7.19 |
| **Wealth quintile** | | | | |
| Poorest | 20.83 | 20.45 | 17.83 | 19.01 |
| Poorer | 21.50 | 20.62 | 19.54 | 20.02 |
| Middle | 19.75 | 22.27 | 21.51 | 21.85 |
| Richer | 20.42 | 21.06 | 22.01 | 21.58 |
| Richest | 17.49 | 15.60 | 19.11 | 17.53 |
| **Religion affiliation** | | | | |
| Christianity | 45.58 | 42.73 | 43.70 | 43.27 |
| Islam | 52.81 | 56.20 | 55.78 | 55.97 |
| Traditional/Others | 1.61 | 1.07 | 0.52 | 0.77 |
| **Parity** | | | | |
| 0 | 6.17 | 3.06 | 2.00 | 2.48 |
| 1 | 13.60 | 14.38 | 11.30 | 12.68 |
| 2–4 | 44.45 | 44.36 | 45.81 | 45.16 |
| 5 and above | 35.78 | 38.20 | 40.89 | 39.68 |
| **Sex of Head of Household** | | | | |
| Male | 89.40 | 89.24 | 89.59 | 66.78 |
| Female | 10.60 | 10.76 | 10.41 | 33.22 |
| **Partner's educational** | | | | |
| No formal education | 36.48 | 35.80 | 31.74 | 33.57 |
| Primary | 22.23 | 19.78 | 15.23 | 17.29 |
| Secondary | 29.22 | 31.63 | 37.09 | 34.62 |
| Tertiary | 12.07 | 12.79 | 15.94 | 14.51 |
| **Experienced child death** | | | | |
| No | 60.58 | 65.75 | 67.63 | 66.78 |
| Yes | 39.42 | 34.25 | 32.37 | 33.22 |
| **Community socioeconomic status** | | | | |
| Low | 51.88 | 53.58 | 49.01 | 31.75 |
| Moderate | 8.01 | 13.99 | 10.90 | 30.98 |
| high | 40.11 | 32.43 | 40.08 | 37.28 |

*(Continued)*

**Table 2.** (Continued)

| Characteristics | Unmet need | | | |
|---|---|---|---|---|
| | 2008 | 2013 | 2018 | Pooled (2008–2018) |
| **Community knowledge of modern contraceptives** | | | | |
| Low | 28.01 | 0.00 | 29.10 | 27.64 |
| Medium | 33.26 | 32.53 | 70.90 | 30.70 |
| High | 38.73 | 67.47 | 0.00 | 41.66 |

zero parity women, female headed household, partner's primary level of education or no formal education and those women who had experienced child death. Other contributors to declined unmet need for family planning between 2008 and 2013 were women whose community socioeconomic status and knowledge of modern contraceptives were classified as higher.

Furthermore, between 2013 and 2018, the proportion of unmet need for family planning increased by 17.33% from 16.10% in 2013 to 18.89% in 2018. Major variation across socio-demographic characteristics and the survey's year shows that the increase is more pronounced among women aged 15–19, 20–24 and 25–29. Regional variations show major increase from the Southern part of the country compared to the Northern part of the country. The major increase in unmet need for family planning were from south East, South West and South South region of the country. Similarly, women who are rural dwellers, those who have attained secondary or tertiary level of education, richer and richest women as well as Christians have contributed immensely to the change recorded in unmet need for family planning between 2013 and 2018. Also, those who have two or more children, those who are from male headed household, who have not experienced child health, whose partners have secondary education and above, whose community socioeconomic status were high and knowledge of modern contraceptives were classified as either low or medium.

## Bivariate results

Chi-square tests were performed at the bivariate level of analysis to assess the associations between unmet needs for family planning and the selected explanatory variables. The results presented in Table 3A indicate that age, region, education, wealth quantile, religion, parity of respondents, sex of household head, partner education, experience of child's death and community literacy are all significant predictors of unmet need for family planning among women of reproductive age between 2008 and 2018 with p values < 0.000. As in the overall percentage changes across the referenced period (2008–2018) for unmet need for family planning (20.21%, 16.10% and 18.89%), there was an initial decline in UNFP between 2008 and 2013, but later increased between 2013 and 2018. This similar pattern occurred across the selected socio-demographic characteristics.

## Multilevel results

**Contextual factors associated with unmet need for family planning in Nigeria.** Table 4 presents the fixed and random effects results of the multilevel mixed effect logistic regression on unmet need for family planning among currently married women (15–49) in Nigeria using the pooled (2008–2018 NDHS) dataset. The results showed that age, educational level, wealth status, religious affiliation, parity, sex of head of household, partner education, region of residence and community socioeconomic status were significant factors associated with unmet need for family planning.

**Table 3. a: Percentage distribution of unmet need for family planning by background characteristics of currently married women (15–49) Nigeria DHS 2008–2018.**

| Variables | Unmet need | | | p value $\chi^2$ |
|---|---|---|---|---|
| | 2008 Yes N (%) 4765 (20.21) | 2013 Yes N (%) 4480 (16.10) | 2018 Yes N (%) 5494 (18.89) | |
| **Age** | p<0.000 | p < 0.000 | p < 0.000 | <0.000 $\chi^2 = 191.32$ |
| 15–19 | 19.05 | 13.05 | 12.17 | |
| 20–24 | 21.43 | 16.56 | 16.10 | |
| 25–29 | 20.41 | 16.81 | 17.05 | |
| 30–34 | 20.10 | 17.15 | 20.83 | |
| 35–39 | 21.29 | 17.61 | 23.14 | |
| 40–44 | 21.39 | 16.81 | 22.38 | |
| 45–49 | 16.19 | 11.54 | 16.55 | |
| **Region** | p < 0.000 | p < 0.000 | p < 0.000 | <0.000 $\chi^2 = 54.39$ |
| North Central | 18.72 | 23.49 | 20.51 | |
| North East | 17.57 | 17.55 | 19.03 | |
| North West | 20.79 | 12.03 | 14.37 | |
| South East | 18.12 | 12.52 | 17.68 | |
| South South | 25.94 | 22.16 | 27.81 | |
| South West | 19.66 | 15.45 | 22.26 | |
| **Education** | p < 0.000 | p < 0.000 | p < 0.000 | <0.000 $\chi^2 = 54.07$ |
| No formal education | 19.24 | 14.86 | 16.79 | |
| Primary | 22.65 | 19.35 | 21.39 | |
| Secondary | 21.58 | 17.28 | 21.23 | |
| Tertiary | 14.62 | 11.72 | 17.15 | |
| **wealth quintile** | p < 0.000 | p < 0.000 | p < 0.000 | <0.000 $\chi^2 = 35.34$ |
| Poorest | 18.35 | 14.27 | 16.30 | |
| Poorer | 20.28 | 15.43 | 17.24 | |
| Middle | 21.84 | 20.02 | 21.10 | |
| Richer | 23.08 | 18.71 | 21.59 | |
| Richest | 18.16 | 12.95 | 18.56 | |
| **Religion affiliation** | p = 0.074 | p < 0.000 | p < 0.000 | <0.0001 $\chi^2 = 39.55$ |
| Christianity | 21.01 | 18.00 | 20.77 | |
| Islam | 19.62 | 14.90 | 17.64 | |
| Others | 18.80 | 16.65 | 18.19 | |
| **Parity** | p < 0.000 | p < 0.000 | p < 0.000 | <0.000 $\chi^2 = 250.99$ |
| 0 | 12.23 | 4.86 | 4.71 | |
| 1 | 17.98 | 15.21 | 14.27 | |
| 2–4 | 19.27 | 15.64 | 18.47 | |
| 5 and above | 25.91 | 21.22 | 25.59 | |
| **Sex of Head of Household** | p = 0.003 | p < 0.000 | p < 0.000 | <0.000 $\chi^2 = 68.93$ |
| Male | 19.91 | 15.71 | 18.38 | |
| Female | 23.12 | 20.24 | 24.69 | |
| **partner's educational** | p = 0.003 | p < 0.000 | p < 0.000 | <0.000 $\chi^2 = 37.79$ |
| No formal education | 18.96 | 14.49 | 17.03 | |
| Primary | 20.86 | 17.22 | 19.35 | |
| Secondary | 21.65 | 18.16 | 20.52 | |
| Tertiary | 19.44 | 15.07 | 18.60 | |
| **Experienced child death** | p = 0.001 | p = 0.523 | p = 031 | <0.0033 $\chi^2 = 8.65$ |
| No | 19.46 | 15.97 | 18.50 | |
| Yes | 21.48 | 16.35 | 19.73 | |
| **Community literacy level** | p < 0.000 | p < 0.000 | p < 0.000 | <0. 0027 $\chi^2 = 5.95$ |
| Low | 17.70 | 13.71 | 15.44 | |
| Medium | 22.02 | 19.87 | 20.19 | |
| High | 20.83 | 15.29 | 21.04 | |
| **Community knowledge of modern contraceptives** | p = 0.002 | p = 043 | P = 0.128 | <0. 0057 $\chi^2 = 5.17$ |
| Low | 19.27 | 15.06 | 18.07 | |
| Medium | 23.76 | 16.65 | 19.24 | |
| High | 20.90 | n.a | 18.89 | |

**Table 4. Multilevel mixed-effect logistic regression on unmet need for family planning among currently married women (15–49) in the Nigeria 2008–2018 DHS dataset.**

| Variables | Null Model AOR (95%C.I) | Model I Individual and Household level AOR (95%C.I) | Model II community level AOR (95%C.I) | Model III Individual/ Household and community level AOR (95%C.I) |
|---|---|---|---|---|
| **Individual and Household level variables** | | | | |
| **Age** | | | | |
| 15–19 (RC) | | 1.00 | | 1.00 |
| 20–24 | | 0.77 (0.68, 0.87)* | | 0.75 (0.67, 0.85)* |
| 25–29 | | 0.60 (0.53, 0.67)* | | 0.58 (0.51, 0.65)* |
| 30–34 | | 0.58 (0.51, 0.66)* | | 0.56 (0.49, 0.63)* |
| 35–39 | | 0.55 (0.48, 0.62)* | | 0.52 (0.46, 0.59)* |
| 40–44 | | 0.48 (0.42, 0.55)* | | 0.46 (0.40, 0.53)* |
| 45–49 | | 0.31 (0.27, 0.36)* | | 0.30 (0.26, 0.34)* |
| **Education** | | | | |
| No education (RC) | | 1.00 | | 1.00 |
| Primary | | 1.16 (1.08, 1.24)* | | 1.09 (1.01, 1.07)** |
| Secondary | | 1.44 (1.33, 1.56)* | | 1.36 (1.26, 1.48)* |
| Tertiary | | 1.26 (1.11, 1.42)* | | 1.21 (1.07, 1.37)** |
| **wealth quintile** | | | | |
| Poorest (RC) | | 1.00 | | 1.00 |
| Poorer | | 1.10 (1.02, 1.18)** | | 1.08 (1.00, 1.16)** |
| Middle | | 1.24 (1.15, 1.34)* | | 1.16 (1.07, 1.26)* |
| Richer | | 1.14 (1.05, 1.25)* | | 1.06 (0.97, 1.16) |
| Richest | | 0.93 (0.84, 1.02) | | 0.85 (0.77,0.95)* |
| **Religion affiliation** | | | | |
| Christianity (RC) | | 1.00 | | 1.00 |
| Islam | | 0.97 (0.91, 1.03) | | 1.67 (1.09, 1.25)* |
| Others | | 0.74 (0.59, 1.93)* | | 0.80 (0.64, 1.00)** |
| **Parity** | | | | |
| 0 (RC) | | 1.00 | | 1.00 |
| 1 | | 3.64 (3.14, 4.22)* | | 3.59 (3.10, 4.16)* |
| 2–4 | | 5.50 (4.75, 6.37)* | | 5.45 (4.71, 6.31)* |
| 5 and above | | 10.73 (9.18, 12.53)* | | 10.69 (9.15, 12.49)* |
| **Sex of head of household** | | | | |
| Male (RC) | | 1.00 | | 1.00 |
| Female | | 1.23 (1.15, 1.33)* | | 1.22 (1.13, 1.31)* |
| **partner's educational** | | | | |
| No education (RC) | | 1.00 | | 1.00 |
| Primary | | 0.97 (0.90, 1.04) | | 0.94 (0.87, 1.01) |
| Secondary | | 1.18 (1.09, 1.27)* | | 1.11 (1.03, 1.20)* |
| Tertiary | | 1.04 (0.96, 1.13)** | | 1.06 (0.97, 1.17) |
| **Experienced child death** | | | | |
| No (RC) | | 1.00 | | 1.00 |
| Yes | | 0.94 (0.89, 0.99)* | | 0.96 (0.92, 1.01) |
| **Region** | | | | |
| North Central (RC) | | | 1.00 | 1.00 |
| North East | | | 0.83 (0.77, 0.90)* | 0.82 (0.75, 0.89)* |
| North West | | | 0.53 (0.49, 0.57)* | 0.53 (0.49, 0.58)* |
| South East | | | 0.53 (0.47, 0.60)* | 0.58 (0.52, 0.66)* |
| South South | | | 0.83 (0.74, 0.92)* | 0.88 (0.78, 0.98)** |
| South West | | | 0.53 (0.47, 0.60)* | 0.60 (0.53, 0.68)* |
| **Community-level Variables** | | | | |
| **Community literacy level** | | | | |
| Low (RC) | | | 1.00 | 1.00 |
| Medium | | | 1.01 (0.92, 1.11) | 0.97 (0.89, 1.07) |
| High | | | 0.90 (0.81, 1.01) | 0.92 (0.82, 1.02) |
| **Community socioeconomic status** | | | | |
| Low (RC) | | | 1.00 | 1.00 |
| Moderate | | | 1.32 (1.19, 1.45)* | 1.25 (1.14, 1.38)* |
| high | | | 2.21 (1.95, 2.49)* | 2.01 (1.79, 2.27)* |

*(Continued)*

**Table 4.** (Continued)

| Variables | Null Model AOR (95%C.I) | Model I Individual and Household level AOR (95%C.I) | Model II community level AOR (95%C.I) | Model III Individual/ Household and community level AOR (95%C.I) |
|---|---|---|---|---|
| **Measure of Variation** | | | | |
| *Random effects* | 0.2472618 | 0.1818451 | 0.1994711 | 0.1695193 |
| Variance | Reference | 26.46 | 19.33 | 31.44 |
| Explained Variance PCV (%) | 6.99 | 5.24 | 5.72 | 4.90 |
| ICC (%) | -30554.46 | -29063.91 | -30283.31 | -28876.53 |
| *Model Fitness* | 61114.92 | 58179.82 | 60590.61 | 57823.06 |
| Log-likelihood | 61142.79 | 58420.98 | 60702.11 | 58147.70 |
| *Model fit Statistics* | | | | |
| AIC | | | | |
| BIC | | | | |

The null model partitions the variance into two component parts, ICC = intraclass correlation coefficient, SE = standard error, PCV = proportional change in variance, MOR = median odds ratio, AIC = Akaike information criterion, AOR = adjusted odds ratio; significance level: $^*p < 0.001$ $^{**}p < 0.01$ $^{***}p < 0.05$.

**The fixed effects.** At the individual level, older women had lower odds of unmet need for family planning than women in the 15–19 years age group. Among the women in the age groups 20–24, 25–29, 30–34, 35–39, 40–44, and 45–49, 25% (OR = 0.75, 95% CI [0.67, 0.85], p < .05), 42% (OR = 0.58, 95% CI [0.51, 0.65] p < .05), 44% (OR = 0.56, 95% CI [0.49, 0.63], p < .05), 48% (OR = 0.52, 95% CI [0.46, 0.59], p < .05), 54% (OR = 0.46, 95% CI [0.40, 0.53], p < .05), and 70% (OR = 0.30, 95% CI [0.26, 0.34], p < .05) had lower odds of having an unmet need for family planning than women in the age group 15–19. The likelihood of an unmet need for family planning was greater for women who had attained secondary education (OR = 1.36, 95% CI [1.26, 1.48], p < .05), tertiary (OR = 1.21, 95% CI [1.07, 1.37], p < .05), or primary education (OR = 1.09, 95% CI [1.01, 1.07], p < .05) than for women with no formal education. The likelihood of unmet need was greater among women who were affiliated with Islam (OR = 1.67, 95% CI [1.09, 1.25], p < .05) than among Christian women. With regard to the wealth quintile, the unmet need for family planning was 1.08, 1.16 and 1.06 times greater among the poorer, middle, and richer women respectively, than among the poorest women. Nevertheless, the richest women were 20% (OR = 0.85, 95% CI [0.77, 0.95, p < .05) less likely to have an unmet need for family planning than the poorest women were. For parity, the odds of having an unmet need for family planning were 10.69 times greater (OR = 10.69, 95% CI [9.15, 12.49, p < .05) for women who had five or more children than for women who had no children. With regard to the sex of the head of household, the odds of having an unmet need for family planning were 21% greater among female-headed households (OR = 1.21, 95% CI [1.14, 1.29], p < .05) than among male headed households. The odds of having an unmet need for family planning were significantly greater by 11% (OR = 1.11, 95% CI [1.03, 1.20], p < .05) among women whose partner had secondary school education than among women whose partner had no formal education.

At the community level, women who were from other geopolitical zones had lower odds of unmet need for family planning than women from the North Central geopolitical zone/region. Compared with those of women from the North Central geopolitical zone, the odds of unmet need for family planning were 47% (OR = 0.53, 95% CI [0.49, 0.58, p < .05) and 42% (OR = 0.58, 95% CI [0.52, 0.66, p < .05), and 40% (OR = 0.60, 95% CI [0.53, 0.68, p < .05) lower for the North west, South East, South West respectively than for women from the North Central geopolitical zone. Additionally, compared with women who lived in communities with low socio-economic status, those who lived in communities with high socio-economic status were 2.01 times (OR = 2.01, 95% CI [1.79, 2.27], p < .05) more likely to have unmet

needs for family planning. In addition, the random effects results, as presented by the ICC and AIC in Table 3 showed variations in the unmet need for family planning. The ICC results (ICC = 4.9%) showed that the individual and household level factors had greater influence on the variation in the unmet need for family planning than did community factors. Additionally, the AIC indicated that the full model had the best goodness of fit.

## Discussion

This study assessed changes in unmet need for family planning and its contextual determinants among married women of reproductive age over a ten-year period and found some salient results. This study showed an improvement in the percentage of respondents who had unmet need for family planning, as the percentage decreased from 20.21% to 16.10% between 2008 and 2013. However, the percentage of respondents who had unmet need for family planning increased from 16.10% in 2013 to 18.89% in 2018. Similar results were reported for unmet need for spacing and limiting, with an initial decline between 2008 and 2013 but a further increase between 2013 and 2018. The proportion of unmet need for family planning across the referenced year by socio-demographic and economic characteristics declined between 2008 and 2013, but increased between 2013 and 2018. This is in line with some prior studies, including a study which reported 20% unmet need for family planning in Africa [37], as well as a DHS report which found 12% and 7% for spacing and limiting, respectively [6]. In Nigeria, unmet need for family planning fluctuates between 14% and 20% in the past 15 years [6]. However, this result contradicts a study conducted in Mali which reported 35.7% unmet need for family planning [38]. There is a need for policy interventions to be strengthened especially because of the slip in unmet need for family planning between 2013 and 2018.

This study revealed that older women had lower odds of having unmet need for family planning than women aged 15–19 years. Several studies have documented an inverse association between maternal age and unmet need for family planning [4,10]. Lack of economic empowerment, cost and availability of family planning services, and preference for ineffective and short-term methods among women in the younger age group are possible reasons for the lower odds of unmet need for family planning. The policy implication is that more long acting and reversible contraceptives should be encouraged among younger women in Nigeria. In addition, younger women would prefer short-term family planning methods to space their next births and are more likely to have achieved their ideal family size. Older women on the other hand might have achieved their fertility goals and tend to prefer long-acting and reversible family planning methods.

The study further revealed that unmet need for family planning increases with increasing level of education. This study is in agreement with previous studies that the unmet need for family planning was greater among women who had completed or attained primary or secondary education than among women without any formal education [4,7]. Educated women are expected to have access to information and be well knowledgeable about modern family planning methods. This stance echoes the findings of studies conducted in Ghana and Ethiopia [7,19]. The plausible reasons might because women with low literacy level are unaware of contraceptive methods. However, other studies revealed a contrary result, as women with tertiary education had a lower odd of unmet need for family planning. Further explanations are rooted in the supposition that educated women do not have deep-rooted misconceptions about modern family planning methods, have positive attitudes towards modern contraception and well-informed about their benefits [10].

The study also revealed a negative relationship between women in the richest wealth category and unmet need for family planning. This finding is in line with previous studies [4,10]

which report an inverse relationship between unmet need for family planning and high household wealth. Lack of access and cost of procuring contraceptive methods among other constraints, faced by the poor women compared to their richest counterparts might explain this relationship. Interventions should make family planning methods cost-free in Nigeria.

Another finding of this study was that religion was a significant factor influencing the overall unmet need for family planning. Women who belonged to Islamic religious groups had greater unmet need for family planning than Christians and other orthodox followers. Religious teaching and the sociocultural issues might explain this relationship. Islamic religion encourages polygamy and large family size and as such, unmet need for family planning tends to be higher. These findings were also reported in previous studies [4,8,11,16] but contradict what others [10,12] reported in Nigeria. This difference might be due to aggregation of the study population. Policies should be focused on women of Islamic faith with a view to reducing unmet need for family planning.

There is a positive and significant relationship between parity and unmet need for family planning. The likelihood of having an unmet need for family planning was greater among women of higher parity. These findings align with previous studies [4,18,39]. This could be attributed to lack of desire to change the practice of high fertility behaviour. This stance contradicts the common knowledge that women who have high parity would limit and space birth because they have attained their desired family size or adhere to harsh economic realities in the country. However, it is possible that this group of women might have not attained their ideal family size and are more likely to higher unmet need for family planning. Interventions should focus on high parity women to increase demand for family planning and in turn reduce unmet need for limiting.

Unmet need for family planning was greater among women living in female-headed households than among women living in male-headed households. This finding is consistent with a previous study [22]. This stance contradicts the gender and power theory which promotes masculinity and arbitrary domination of women in decision-making power, especially fertility behaviour.

With regard to husbands' educational level, the study revealed a similar pattern of association with women's education and unmet need for family planning. This finding is in line with some previous studies [39,40]. It is expected that in a patrilineal society, educated men will understand the risk of having a large family size compared to men without formal education.

This study also revealed regional disparities in the unmet need for family planning. This finding is consistent with previous findings [4,10,12,20]. Although difficult to explain, the disparities in region/geo-political zones might be mainly due to culture, socioeconomic inequalities, religion, lack of access to and attitudinal resistance to certain contraceptive methods in some regions/geo-political zones compared to others. The study revealed that North Central women had higher unmet need for family planning compared to other regions.

With respect to community socioeconomic status, the study showed that women who lived in communities with high socio-economic status had greater unmet need for family planning than did their counterparts who lived in communities with low socio-economic status. Some studies reported similar results [20,41,42]. The information received from community-based family planning initiatives, friends, and relatives coupled with contraceptive behaviour in high socio-economic communities might have influenced unmet need for family planning. Community characteristics are particularly important for improving women's contraceptive behaviour because family planning information is provided within women's social networks. This has implications on fertility behaviour, and this supports the need for more community-based family planning initiatives to improve access to modern contraception in the communities.

The study revealed that the community-level factors had little influence on the variation in unmet need for family planning in Nigeria.

## Strengths and limitations of the study

The NDHS dataset used for this study contains high-quality representative data at the national and subnational levels. It consists of population-based data that are in the public domain. The generalization of the findings, replication and verification of the study are major strengths. The use of multilevel mixed effects logistic regression models makes the findings more generalizable. The use of the pooled NDHS datasets (2008, 2013 and 2018) enabled us to measure the total prevalence of unmet need for family planning for the three surveys. This study is a novel contribution to knowledge base owing to its investigation of contextual factors responsible for changes in unmet need for family planning. However, due to the cross-sectional nature of the datasets, causality between variables cannot be established. In addition to this, the DHS collected self-reported retrospective information. Self-reported information was collected five years prior to the survey. Thus, recall bias might be a weakness of this study.

## Conclusion

This study examined the changes and contextual determinants of unmet need for family planning in Nigeria. The findings showed that the percentage of respondents who had unmet need for family planning dropped between 2008 and 2013 but later increased between 2013 and 2018. This showed a slip in the improvement achieved regarding unmet need for family planning reduction. Age, educational level, wealth status, religious affiliation, parity, head of household sex, partner educational level, region of residence and community socioeconomic status were also found to be significant factors associated with the unmet need for family planning in Nigeria. The study concluded that unmet need for family planning is high and community-level factors had little variation (odds ratios and percentages) in the unmet need for family planning compared to household and individual level factors which constantly declined between 2008 and 2013 and later increased in 2018. The study suggested that the Ministry of Health and all stakeholders in family planning in Nigeria must focus on individual and household factors in order to improve the use of family planning services across the country. Future studies should focus on spatial distribution and contextual determinants of unmet need for family planning among women of reproductive age in Nigeria or in some selected countries in sub-Saharan Africa.

## Author Contributions

**Conceptualization:** Funmilola Folasade Oyinlola.

**Data curation:** Funmilola Folasade Oyinlola, Joseph Ayodeji Kupoluyi, Olufemi Mayowa Adetutu.

**Formal analysis:** Funmilola Folasade Oyinlola, Joseph Ayodeji Kupoluyi, Olufemi Mayowa Adetutu.

**Investigation:** Funmilola Folasade Oyinlola, Joseph Ayodeji Kupoluyi.

**Methodology:** Funmilola Folasade Oyinlola, Joseph Ayodeji Kupoluyi, Olufemi Mayowa Adetutu.

**Project administration:** Funmilola Folasade Oyinlola, Joseph Ayodeji Kupoluyi, Olufemi Mayowa Adetutu.

**Resources:** Funmilola Folasade Oyinlola, Joseph Ayodeji Kupoluyi.

**Software:** Joseph Ayodeji Kupoluyi.

**Supervision:** Joseph Ayodeji Kupoluyi.

**Writing – original draft:** Olufemi Mayowa Adetutu.

**Writing – review & editing:** Joseph Ayodeji Kupoluyi, Olufemi Mayowa Adetutu.

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
