## [Decision Letter · Decision Letter 0]

10 Mar 2024

PONE-D-24-04879Changes in unmet need for family planning among women of reproductive age in Nigeria: a multilevel analysis of a ten-year DHS wavePLOS ONE

Dear Dr. OYINLOLA,

Thank you for submitting your manuscript to PLOS ONE. After careful consideration, we feel that it has merit but does not fully meet PLOS ONE’s publication criteria as it currently stands. Therefore, we invite you to submit a revised version of the manuscript that addresses the points raised during the review process.

We look forward to receiving your revised manuscript.

Kind regards,

Obasanjo Afolabi Bolarinwa, Masters

Academic Editor

PLOS ONE

Journal Requirements:

Reviewers' comments:

Reviewer's Responses to Questions

**Comments to the Author**

1. Is the manuscript technically sound, and do the data support the conclusions?

Reviewer #1: Partly

Reviewer #2: Partly

Reviewer #3: Partly

2. Has the statistical analysis been performed appropriately and rigorously? 

Reviewer #1: No

Reviewer #2: Yes

Reviewer #3: No

3. Have the authors made all data underlying the findings in their manuscript fully available?

Reviewer #1: Yes

Reviewer #2: No

Reviewer #3: Yes

4. Is the manuscript presented in an intelligible fashion and written in standard English?

Reviewer #1: Yes

Reviewer #2: No

Reviewer #3: Yes

5. Review Comments to the Author

Reviewer #1: The paper ‘Changes in unmet need for family planning among women of reproductive age in Nigeria: a multilevel analysis of a ten-year DHS wave’ uses secondary data from the Nigeria Demographic and Health Survey to outline the changes in family planning among women over time. It attempts to use secondary data to show changes in the unmet need for family planning. The researchers use trend analysis and multilevel analysis where they combined all the datasets across 3 waves of DHS (2008, 2013 and 2018) DHS studies.

The paper falls short of its objectives. It does very little to show the changes across the period in unmet need for family planning. While it shows progress in unmet need using a trend analysis, this does not delve into the changes and potentials that could be responsible for these changes.

The researchers however, investigated the factors that could influence the unmet need for family planning. They start by making an effort to use Chi-square from bivariate analysis to make a statement on the observation. However, this statement is confusing. ‘Table 2 indicate that age, region, education, wealth quantile, religion, parity of respondents, sex of household head, partner education, experience of child’s death and community literacy are all significant predictors of unmet needs for family planning among women of reproductive age between 2008 and 2018 with p values < 0.000.’ The only thing Chisquare would tell us is that the proportions in each group being compared are different. The values could be swinging and going up and down and this does not tell in any way if they are predictors of unmet need for family planning.

The researchers note that their study is a cross-sectional study. It has been suggested in the annotated document that the researchers should recast that the study used secondary data that had been collected through a cross-sectional survey. They didn’t design the study and cannot now change the design.

For factors that are associated with unmet need for family planning, the researchers used an appropriate method. However, the study is noted to be about the changes in unmet need which this investigation failed to elaborate on. There are several studies that have investigated the factors influencing unmet need for family planning and one does not immediately see the added value that this analysis brings to the table as these same predictors have been identified in previous studies.

Minor Corrections

Introduction

This statement: ‘Promotion of use of modern contraception has been mainly advocated by non-governmental, international and community-based organisations.’ – Would be good to give a reference for such statement. Otherwise, it is your opinion. I am certain several governments have made policy commitments towards control of population explosion. China is an extreme one. So, your statement might not be absolutely correct.

Meanwhile, government interventions from international organizations and other stakeholders aimed at reducing high unmet need for family planning among women by making family planning methods available and accessible have yielded little progress in Nigeria [13, 14]. - Can you explain better what you mean by this statement? You had stated earlier that government wasn't doing much. You might want to revise that statement.

For instance, in Ghana the importance of studies on unmet need for improving family planning uptake was established [14, 15, 16]. – Established what?

In a study, a prevalence of 30% was found among married Ghanaian women [15]. – Sentence can be better cast. 30% of what?

A cross-sectional research study design using secondary data from three rounds of the Nigeria Demographic and Health Survey (NDHS) conducted in 2008, 2013, and 2018 was employed in this study. The women of reproductive age (15–49 years) in the surveys were asked retrospective questions covering the 5 years preceding the survey. - Did you use a cross sectional study design? Or was the secondary data you used collected through cross-sectional means? Your statement is as though you designed the study. Please revise appropriately.

Exclusion and Inclusion

The revised definition of unmet need for family planning [17, 25] was applied to estimate total

unmet need (spacing and limiting). - Good to explicitly provide this definition here.

‘women who were not using contraception, who were pregnant or amenorrheic, or whose pregnancy or birth was mistimed or unwanted;’ – Between amenorrhoeic and Whose should have been an ‘AND’ operator from the schema diagram that you provided.

Explanatory variable

The individual level variables were as follows: age – This should be age group

Statistical Analysis

These three sets of DHS datasets were used separately to examine the regional trends in unmet need for family planning, while the data were pooled to examine the contextual correlates of unmet family planning needs in Nigeria. - Why pool the data if you believe it should be changing from year to year? Or what is the rationale for the trend analysis and the title of ‘Change’?

Reviewer #2: Dear editorial team,

Thank you for inviting me to review this fascinating title on the global agenda of women's health. Dear authors, I appreciate your efforts in coming up with an interesting title, although it does require substantial revision. Below, I have provided section-by-section review comments that I highly recommend you address point by point.

General comments

Introduction:

The introduction lacks clarity and fails to provide a strong background for the study. Please consider rephrasing it to clearly outline the purpose and significance of the research.

Literature Review:

The literature review is comprehensive but lacks recent references. Please update it with more current sources to ensure the study is up to date.

Methodology:

The methodology section needs further elaboration. Please provide more details on the sample size, data collection methods, and statistical analysis techniques used.

Results:

The results section is well-presented and provides valuable insights. However, it would be beneficial to include more graphical representations, such as charts or tables, to enhance the clarity of the findings.

Discussion:

The discussion section needs to be more focused and structured. Please organize the key findings and relate them back to the research objectives. Additionally, consider discussing the limitations of the study and suggesting areas for future research.

Conclusion:

The conclusion should be more concise and summarize the main findings of the study. Avoid introducing new information in this section.

Language and Writing Style:

The overall language and writing style of the manuscript need improvement. Please ensure that the text is clear, concise, and free from grammatical errors. Consider seeking assistance from a professional editor to enhance the readability of the paper.

Specific comments

Abstract

1. Methods and materials could revised as methods only

2. You study about unmet need for family planning while you concluded about family planning, since unmet need isnot equal with family planning

Introduction

1. It is good while some grammatical errors there and

2. There is also study conducted on unmet need for family planning while you did not mentioned it (Solanke et al.2022,adebowale et al,2023 based on 2018 DHS…)

3. Since DHS data is open access you did not assure it is whether conducted or not similar study

4. There is study conducted based on Nigeria DHS 2018 which is recent data and important for policy and program, if this study conducted what is the important of studying on same title?

5. Trends of unmet need conducted by Oginni et al. 2015(based on DHS 2008-2013) and Adebowale et al. 2023( based on 2018 DHS) if so what new finding is added this study?

Methods

1. How to manage missing values

2. Detailed analysis of the data not described

3. Description of independent variables mention in tables

4. What about clustering effect

5. Random effect and fixed effect, is two equally used how?

Results

1. What does mean Univariate Results? In table descriptive statistics/sociodemographic characteristic

2. Missing value not reported, why? Since this is secondary data missed data is expected

3. You use fixed effect model, how? This is country level data so how fixed effect model used?

4. The ICC value decreased what does mean?

Discussion

1. The implication not included please include in discussion

2. Comparison with previous literatures not well discussed

3. Include national level reports of unmet need during discussion

4. Religion is one of the factors, how did you recommended for programs and policy makers

5. You calculated Odds ratio but on discussion it says there is a positive, how?

Limitation

1. What about missed value, using secondary data, behavioral factors?

Reviewer #3: The manuscript focused on changes of unmet need among women of reproductive age in Nigeria.

However, find below some observation

Abstract: The objective states that the study focused on unmet need of reproductive women in Nigeria. However, under the "Material and Method" section in the abstract, it was written that the study examined the trend of unmet need. This is confusing. Kindly state clearly the objective of the study to enable readers follow through.

Introduction

The study does not show the gaps in knowledge that necessitate the present study. The authors stated that between 2015 and 2018, an estimated 210 million pregnancies occurred. Kindly state the source of this information. Paragraph 3 and 4 is clumsy and not coherent. The sentences in the introduction are too long without precision. Please use short sentence. There is no clear direction of the problem in the introduction as merely outlining statistics does not imply problem. Kindly show the pain that gave birth to the study. You could commence from the global scale to the region (Africa) and national (Nigeria). The introduction does not flow systematically.

Method

This is a standard secondary data and most of the details outline were not needed. Moreso, the authors stated similar studies were the method and dataset had been used. Brevity is power

Result

state the prevalence of unmet need per cohort of DHS. What is responsible for variation of the prevalence?

Discussion

summarise key result with reference to the objective of the study

The context of the result is missing. This is expected based on the objective of the study stated at the abstract section

Can this result be generalise? Please state reasons

What is the policy implication of the study

Argument in this section is not coherent

Conclusion

Where is the novelty of the study?

6. PLOS authors have the option to publish the peer review history of their article (what does this mean?). If published, this will include your full peer review and any attached files.

Reviewer #1: No

Reviewer #2: No

Reviewer #3: **Yes: **chukwudeh Okechukwu Stephen

---

## [Author Response · Author response to Decision Letter 0]

26 May 2024

The comments by the reviewers have been incorporated.

---

## [Decision Letter · Decision Letter 1]

25 Jun 2024

Changes in unmet need for family planning among women of reproductive age in Nigeria: a multilevel analysis of a ten-year DHS wave

PONE-D-24-04879R1

Dear Dr. Oyinloye,

We’re pleased to inform you that your manuscript has been judged scientifically suitable for publication and will be formally accepted for publication once it meets all outstanding technical requirements.

Kind regards,

Obasanjo Bolarinwa

Academic Editor

PLOS ONE

Additional Editor Comments (optional):

All comments have been addressed.

Reviewers' comments:

Reviewer's Responses to Questions

**Comments to the Author**

1. If the authors have adequately addressed your comments raised in a previous round of review and you feel that this manuscript is now acceptable for publication, you may indicate that here to bypass the “Comments to the Author” section, enter your conflict of interest statement in the “Confidential to Editor” section, and submit your "Accept" recommendation.

Reviewer #3: All comments have been addressed

Reviewer #4: All comments have been addressed

2. Is the manuscript technically sound, and do the data support the conclusions?

Reviewer #3: Yes

Reviewer #4: Yes

3. Has the statistical analysis been performed appropriately and rigorously? 

Reviewer #3: I Don't Know

Reviewer #4: Yes

4. Have the authors made all data underlying the findings in their manuscript fully available?

Reviewer #3: Yes

Reviewer #4: Yes

5. Is the manuscript presented in an intelligible fashion and written in standard English?

Reviewer #3: Yes

Reviewer #4: Yes

6. Review Comments to the Author

Reviewer #3: The study examined changes in unmet need for family planning and its contextual determinants among married women of reproductive age over a ten year period. Socio-economic and demographic factors were found to significantly influence unmet need for family planning in Nigeria. Household and individual factors were implicated for the fluctuation of unmet needs for family planning. The abstract, introduction, method, result, discussion and conclusion has been improved upon.

Please remove the bracket in line 3 in the abstract written as ''The study used three (DHS) conducted over a ten....Rather, write the demographic and health survey (DHS). This is the first usage in the article.

Arrange ''key works'' alphabetically

Reviewer #4: It is fine. Most of information is acceptable. Analytical section has been improving-particularly interpretation of their results. Besides, discussion is fine and related to their main ideas. So, I agree to accept.

7. PLOS authors have the option to publish the peer review history of their article (what does this mean?). If published, this will include your full peer review and any attached files.

Reviewer #3: **Yes: **Chukwudeh Okechukwu Stephen

Reviewer #4: **Yes: **Yothin Sawangdee

---

## [Editor Report · Acceptance letter]

25 Jul 2024

PONE-D-24-04879R1 

PLOS ONE

Dear Dr. Oyinlola, 

I'm pleased to inform you that your manuscript has been deemed suitable for publication in PLOS ONE. Congratulations! Your manuscript is now being handed over to our production team.

Kind regards, 

on behalf of

Mr Obasanjo Bolarinwa 

Academic Editor

PLOS ONE